# Search for New Allergens in *Lolium perenne* Pollen Growing under Different Air Pollution Conditions by Comparative Transcriptome Study

**DOI:** 10.3390/plants9111507

**Published:** 2020-11-06

**Authors:** Jose Antonio Lucas, Enrique Gutierrez-Albanchez, Teresa Alfaya, Francisco Feo Brito, Francisco Javier Gutierrez-Mañero

**Affiliations:** 1Plant Physiology, Pharmaceutical and Health Sciences Department, Faculty of Pharmacy, Universidad San Pablo-CEU Universities, 28668 Boadilla del Monte, Spain; diridbiol@biobab.net (E.G.-A.); jgutierrez.fcex@ceu.es (F.J.G.-M.); 2Allergy Section, General Hospital, 13001 Ciudad Real, Spain; talfaya@sescam.jccm.es (T.A.); feobrito1@gmail.com (F.F.B.)

**Keywords:** air pollution, allergen, *Lolium perenne*, pollen, transcriptome

## Abstract

The relationship between air pollution and the allergenic capacity of pollen is widely accepted, with allergenicity being directly related to air pollution. To our knowledge, this is the first study comparing the differential expression of *Lolium perenne* pollen genes by RNAseq, in two wild populations with different levels of air pollution. The objective is to search for proteins that are expressed differentially in both situations and to establish a relationship with increased allergenic capacity. Two populations of *L. perenne* (Madrid and Ciudad Real) have been studied in two consecutive years, under the rationale that overexpressed genes in Madrid, with higher levels of NO_2_ and SO_2_, could be a cause for their greater allergenic capacity. Heat shock proteins (HSP), glycoside hydrolases, proteins with leucin-rich repeat motifs, and proteins with EF-HAND motifs were consistently overexpressed in Madrid pollen in the two years studied. Interestingly, some genes were overexpressed only in one of the years studied, such as pectinesterases in the first year, and lipid transfer proteins (LTPs) and thaumatin in the second. Despite the fact that the potential of all these proteins in relation to possible allergies has been reported, this is the first time they are cited as possible allergens of *L. perenne*. The results found can contribute decisively to the knowledge of the allergens of *L. perenne* and their relationship with atmospheric pollution, and to the development of much more effective vaccines.

## 1. Introduction

Most immunoglobulin E (IgE)-mediated allergies are caused by the plant’s allergens, which may cause different symptoms such as rhinoconjunctivitis, edema, urticarial, asthma, and anaphylaxis [1]. Nowadays, the incidence of pollen allergy is undergoing a striking increase, with pollen allergens being the main cause among people with perennial allergic rhinitis [2,3]. Allergies and hypersensitive responses that are initiated by specific immunologic mechanisms triggered by pollen and other allergens, constitute one of the major health issues in modern societies [4].

The relationship between air pollution and the increased allergenic capacity of pollen is a widely accepted fact [5,6,7,8,9,10,11]. The relationship between the allergenic capacity of pollen, the degree of air pollution, and the physiological status of the plant has been recently demonstrated [12], revealing that plants growing under a higher atmospheric pollution had a lower photosynthetic efficiency, with altered ROS scavenging systems resulting in a greater degree of oxidative stress, higher H_2_O_2_ concentration, and enhanced NADPH oxidase activity in pollen. These two factors (H_2_O_2_ concentration and NADPH oxidase) are considered as very relevant in the increase of the allergenic capacity of pollen [12].

Pollen from different types of plants has been found to trigger allergic reactions [13]. Up to eleven groups of grass pollen allergens have been identified for its ability to elicit a specific IgE response [14]. Among them, pollen of the Poaceae family are the main source allergens as outstanding *Poa pratensis* (Kentucky bluegrass), *Phleum pratense,* and *L. perenne* (perennial ryegrass). [15]. The pollen of perennial ryegrass is the major cause of allergic diseases [16,17,18]. 

On the other hand, according to clinical data collected by allergists consultations through the usual skin tests, the true extent of the problem is not evidenced, as there are many more cases of pollen allergies. To date, six pollen allergens of *L. perenne* have been described and it is assumed that they are always present in pollen of this species, regardless of the place of collection. This is certainly one of the problems, since pollen used in skin tests rarely comes from the place where the patient lives. Furthermore, it is necessary to investigate new allergens, since in many cases vaccines do not have the desired effect, probably because they are not being manufactured with the appropriate allergens (Dr. F. Feo and Dra. T. Alfaya personal communication).

Plants have an inducible adaptative metabolism known as secondary metabolism, through which they are able to adapt to both biotic and abiotic stresses. This metabolism is capable of synthesizing a plethora of molecules of different nature that only appear under stress [19]. The best known and most studied are molecules related to the defense against pathogens. Some of them have demonstrated their allergenic capacity in some plant’s pollen, such as lipid transfer proteins (PR 14) or thaumatin (PR 5) [20]. As discussed above, the relationship between air pollution and the greater allergenic capacity of some pollen has already been demonstrated, which is surely related to the synthesis of molecules induced by these environmental conditions, probably allergenic molecules not yet described. 

Pollen used in this work was collected from two cities in central Spain (Madrid and Ciudad Real), near heavy traffic areas in both cases. However, pollution levels, especially nitrogen oxides and sulfur oxides, were higher in Madrid throughout the plant’s vegetative and reproductive period [12]. The pollen was collected from the plants before leaving the anthers.

Once the relationship of air pollution with a greater allergenic capacity of pollen had been demonstrated, the objective of this work was to identify overexpressed genes in the high pollution area (Madrid) as compared to the low pollution area (Ciudad Real, considered as control) by means of a transcriptomic analysis, with the secondary aim to find new potential allergens in the pollen of *L. perenne*. In order to achieve these objectives, the gene differential expression of pollen from both localities was studied using the RNAseq technique, in two consecutive years, placing a special emphasis on those genes that were overexpressed in the Madrid pollen, and that could be responsible for its greater allergenic capacity. 

## 2. Results

To construct a transcriptome database, six mRNA libraries were generated in each collection moment (May 2017 and 2018) by Illumina sequencing, three from each population of *L. perenne* from Ciudad Real and three from Madrid.

Table 1 summarizes the mapping results. Among the 52 million readings (on average with 101 bp read length) obtained in each sample, 53% of them were mapped. Uniformity between the samples, indicated that samples were comparable. Usually, it is possible to align 60–90% of the reads to the reference genome. However, this data depends upon the quality of the sample and the coverage of the reference genome. The highest percentages are obtained with very well-curated model organism genomes. 

A total of 666,180 genes were identified in 2017 after sequencing, mapping alignment, normalized expression, and differential expression, but 550,285 genes in 2017 were not available for the study, since the fold change and *p* adjust value were not obtained in the normalized expression analysis. A total of 115,895 genes were available for the genetic expression analysis, of which 107,599 had a *p* adjust value over 0.05. When the pollen of the two cities was compared, the expression pattern showed that in 2017, 366 genes were common to both cities (expression without significant differences), and 2443 genes were significantly overexpressed in the pollen from Madrid, while 5487 genes were significantly overexpressed in Ciudad Real (Figure 1). 

A total of 666,181 genes were identified in 2018 after sequencing, mapping alignment, normalized expression, and differential expression, but 585,368 genes were not available for the study, since the fold change and *p* adjust value were not obtained in the normalized expression analysis. A total of 80,813 genes resulted as available for the genetic expression analysis, of which 69,239 had a *p* adjust value over 0.05. When the pollen of the two cities was compared, the expression pattern showed that in 2018, 163 genes were common to both cities (expression without significant differences), and 7568 genes were significantly overexpressed in the pollen from Madrid, while 3823 genes were significantly overexpressed in Ciudad Real (Figure 2).

Table 2 shows the genes that have been identified with some description in any of the Gene Ontology (GO), KEGG Orthology (KO), Pfam, and Clusters of Orthologous Groups (COG) databases, among the 50 most overexpressed (based in the fold change value) in Madrid and Ciudad Real in both years.

In both years, the highest number of genes identified corresponded to genes overexpressed in the pollen from Madrid, eight out of 50 in 2017 and 25 out of 50 in 2018 compared to two and eight in Ciudad Real, respectively.

In Ciudad Real, overexpressed genes were related to the primary metabolism, none of which was involved in routes of secondary adaptive metabolism by which potential allergenic molecules were synthesized. Conversely, among the genes overexpressed in the Madrid pollen some were involved in secondary metabolism, especially in 2018. In 2017, genes involved in the shikimic acid pathway, in the phenylpropanoid pathway, and heat shock proteins of family 20 (HSP 20) were overexpressed. In 2018, almost all overexpressed genes were related to secondary metabolism. There were 14 isoforms of the heat shock protein of 20 family, two isoforms of allergen 1 from *Betula verrucosa* (Bet v 1), and two lipid transport proteins (LTPs). All of them have been described as allergens.

The GO analysis (Figure 3 and Figure 4) shows the most abundant genes grouped in three categories: Cellular components, molecular function, and biological process. Common to both years, the most abundant genes corresponded to “cell parts”, “organelle” and “cell” subcategories within “cellular components” category, “catalytic activity” and “binding” subcategories within “molecular function” category, and “metabolic process” and “cellular process” within “biological process” category and specific to 2018, “biological regulation” and “response to stimulus” within “biological process” category (Figure 4).

Among the upregulated genes in Madrid samples, ten showed isoforms in which overexpression was different depending on the year (Table 3); the number of isoforms from the six genes overexpressed in both years (HSP, glycoside hydrolase, Leucin rich repeat, EF hand family, pollen allergy, and coifilin), only in 2017 (Pectinesterase and serpin) and only in 2018 (lipid transfer protein and thaumatin) are indicated in the table. To make this table, the classification of functions proposed by the Pfam database (Appendix A) has been taken into account. Only the genes overexpressed in Madrid have been taken into account since the greater allergenic capacity of this pollen with respect to that of Ciudad Real has already been shown, as well as its relationship with the atmospheric pollution. Therefore, the genes overexpressed in Madrid are candidates to be responsible for this greater allergenic capacity. 

## 3. Discussion

Nowadays, allergic diseases have become a pandemic health problem. Among them, pollen allergies are considered the most important [21]. Some studies showed that most of the patients sensitized to pollen allergens have perennial allergic rhinitis [21,22]. Moreover, it is demonstrated that these diseases appear to be more prevalent in industrialized countries, and the incidence seems to be higher in polluted areas, especially areas with heavy traffic [4,12,23]. 

Some studies have shown the existence of an in situ allergic response in patients with negative skin prick test (SPT) results and undetectable IgE in the serum [24]. This clinical entity, known as local allergic rhinitis (LAR) [25], is considered a new phenotype of allergic rhinitis (AR) that must be differentiated from nonallergic rhinitis [26,27]. This misunderstanding could be related to several facts: (i) In most cases, pollen to which patients are exposed is not the same as that used in skin prick tests; (ii) the number of allergens involved in the allergic processes may be greater than what has been described so far; and (iii) probably some of the allergens are only expressed upon specific physiological conditions of plants and, among which is the degree of atmospheric pollution [12].

The International Union of Immunological Societies (WHO/IUIS) Allergen Nomenclature Sub-committee (http://www.allergen.org/search.php?allergensource=Lolium+perenne) establishes six allergens in *Lolium perenne* (Lol p 1, Lol p2, Lol p 3, Lol p 4, Lol p 5, and Lol p 11). The first three are expansins, proteins specialized in pollination, with its role being to weaken the cell wall during the development of the pollen tube.

The study of differential expression by RNAseq for two consecutive years in the *L. perenne* pollen, in two cities with different levels of atmospheric pollution intends to identify new allergens of *L. perenne* in order to improve the immunogenic therapy, making it more effective. At the same time, determining allergens related to higher air pollution levels could shed some light to explain why many patients with a negative skin test, still show local signs of allergy.

Genes encoded for *L.perenne* pollen allergen 1 (Lol p 1), were overexpressed in Madrid samples, but they were by no means the most overexpressed. The most overexpressed genes in the Madrid pollen in the two years studied were heat shock proteins (HSP), specifically with a molecular weight of 20, 70, and 90 (Table 2). Most of them are from the chaperones family, and their mission is to refold damaged proteins after stressful situations [28]. They were first described in relation to heat stress situations, but they have been reported in many other stress situations. Some of these proteins have been described as allergy-causing agents in fungi, mites, chestnut (Cas s 9 is an HSP20), and hazelnut pollen (Cor a 10 is an HSP70) [29]. However, to date, the allergenicity of *L. perenne* pollen has not been related to HSP. 

Other highly overexpressed genes in both years of study were those related to glycoside hydrolases (Table 2) (EC 3.2.1.), a widespread group of enzymes that hydrolyze the glycosidic bond between two or more carbohydrates, with a group of 100 different families according to the sequence similarity [30,31,32]. Family 17 showed the highest levels of expression in both years. Glycoside hydrolase family 17 includes enzymes with several activities such as endo-1,3-beta-glucosidase (EC3.2.1.39), lichenase (EC 3.2.1.73), or exo-1,3-glucanase (EC 3.2.1.58). Currently, these enzymes have only been found in plants and in fungi. Some glucanases from plants have been described as allergens, i.e., in *Hevea brasiliensis* latex [32], olive pollen [33], and plant foods [34]. However, there are no bibliographic references to these enzymes as putative allergens in *L. perenne.*

Leucine rich repeats (LRR) genes have also been detected to be overexpressed in pollen from Madrid in the two years studied. LRR are repeated sequences present in a number of proteins with diverse functions and cellular locations. These repeated sequences are usually involved in protein-protein interactions. LRR domains are composed of beta-alpha units that form curved horseshoe structures with a parallel beta sheet on the concave side and mostly helical elements on the convex side. LRR domains are often flanked by cysteine rich domains [35,36]. Nowadays, an LRR-containing protein from wheat was found by screening a phage display wheat cDNA library with wheat allergic patients’ IgE [37], but there is no bibliographic reference on *L. perenne*.

Genes of proteins with EF-HAND motifs were also overexpressed in the Madrid pollen in the two years studied. The EF-hand-containing proteins actively bind to Ca^2+^ and chelate the cytosolic calcium to regulate calcium homeostasis [38]. The major EF-hand containing proteins are calcium dependent protein kinases (CDPKs/CPKs), calcineurin B-like (CBL), calmodulin-like proteins (CMLs), and calmodulins (CaMs). Allergens of this type have been described in the pollen of many different plants as *Alnus glutinosa* (Aln g 4), *Brassica napus* (Bra n polcalcin), *Chenopodium album* (Che a 3), *Olea europaea* (Ole e 3 and Ole e 8), etc. However, there is no reference to *L. perenne*.

We have found that some genes overexpressed in Madrid in the first or second year of study, but not in both. During the first year, genes that code for pectinesterases (seven isoforms) were overexpressed. These enzymes play an important role in the cell wall metabolism during fruit ripening [39]. Sal k 1 from the *Salsola kali* pollen was shown to be a major allergen [40].

During the second year of study, seven genes that code for isoenzymes of lipid transfer proteins (LTPs) and six isoenzymes of thaumatin were overexpressed in Madrid. Both are described as pathogenesis-related (PR) proteins and have a reputation for their allergenic capacity. LPTs are from family 14 (PR 14) and thaumatins belong to family 5 (PR 5). To date, the International Union of Immunological Societies Allergen Nomenclature Subcommittee has reported 39 allergenic LTPs from vegetables (*n* = 7), pollen of trees and weeds (*n* = 9), fruits (*n* = 18), nuts and seeds (*n* = 4), as well as latex (*n* = 1) [41]. Thaumatin-like proteins (TLPs) have been known for years as the main allergens in some fruits and pollen, such as allergen 3 from *Cupressus arizonica* (Cup a 3) [20]. There is no report relating the allergenicity of *L. perenne* pollen neither for LPTs nor for TLPs. 

## 4. Material and Methods

### 4.1. Pollen Used in the Study

Pollen from *L. perenne* plants growing in natural conditions in Ciudad Real and Madrid cities was used for the experiments. Pollen was collected from plants in the maximum pollen production period (mid of May 2017 and 2018). Three populations of *L. perenne* separated between 50 and 100 m were selected. Plants of each population were harvested and each one constituted a replicate. Pollen was recollected by the company Iberpolen S.L. (Alcala la Real, Jaen, Spain). 

### 4.2. RNA Library Assembly

Before the RNA library assembly, ribosomal RNA was removed. This was performed with the Ribo-Zero rRNA kit removal kit. The TruSeq Stranded Total RNA library Prep kit was used to generate the libraries of RNA. First of all, 2 μg of total RNA (RIN > 9) libraries, were sequenced using a HiSeq2500 instrument (Illumina Inc, San Diego, CA, USA). Sequenced readings were paired-end with a length of 101 bp reading performed in six samples (three from Madrid and three from Ciudad Real). The estimated coverage was around 59 million reads per sample (one lane). Library generation and RNA sequencing was done at Sistemas Genómicos S.L. (Valencia, Spain) following the manufacturer’s instructions.

### 4.3. RNA Transcriptomics Analysis

The FastQC v0.11.4 tool was used to check the quality control of the raw data. Then, the raw paired-end reads were mapped against the “*Lolium perenne*” ASM173568v1 genome provided by the NCBI database using the Tophat2 2.1.0 algorithm [42]. Insufficient quality reads (phred score < 5) were eliminated using Samtools 1.2 [43] and Picard Tools 2.12.1. Then, the GC distribution (i.e., the proportion of guanine and cytosine bp along the reads) was assessed, this should have a desired distribution between 40–60%. Moreover, to confirm that our sequencing contained a small proportion of duplicates, the distribution of duplicates (quality of sequencing indicator) were evaluated. Expression levels were calculated using the HTSeq [44]. This method employs unique reads for the estimation of gene expression and filters the multi-mapped reads. Differential expression analysis between conditions was assessed using DESeq2 [45]. Finally, we selected differentially expressed genes with a p-value adjusted by FDR < 0.05 and a fold change of at least 1.5 [46]. The DEG analysis between pollen from Madrid and pollen from Ciudad Real was done by using statistical packages designed by Python and R. using the DESeq2 algorithm [45]. By applying a differential negative binomial distribution for the statistics significance [44], we identified the genes that were differentially expressed. We considered as differently expressed genes those with a FC value below −1.5 or higher than 1.5 and with a *p*-value (Padj) corrected by FDR ≤ 0.05 to avoid the identification of false positives across the differential expression data.

### 4.4. Functional Enrichment Analysis

The gene category enrichment analysis was performed by comparing the differentially expressed genes to the Uniprot database by using Blastx and setting an e-value of 0.01 and a minimum of 40% of the protein length/transcript ratio. With the obtained terms, an over representation test was performed using an in-house R Script developed at Sistemas Genómicos (Valencia, Spain). The graphical plotting of DEGs distribution within GO categories was performed at http://wego.genomics.org.cn/.

## 5. Conclusions

In conclusion, the results obtained in this work can be very useful, since we have described genes that code for some overexpressed proteins in conditions of higher air pollution with potential allergenic capacity for the first time in *L. perenne*. These proteins have to be synthesized by heterologous cloning before their allergenicity can be checked with skin tests. In the case of positive responses, our contribution to the knowledge of allergens of *L. perenne* and their relationship with atmospheric pollution would be confirmed. On the other hand, this would contribute to the development of much more effective vaccines, probably solving the problem of an allergic response in patients with a negative skin prick test (SPT).

## Figures and Tables

**Figure 1 plants-09-01507-f001:**
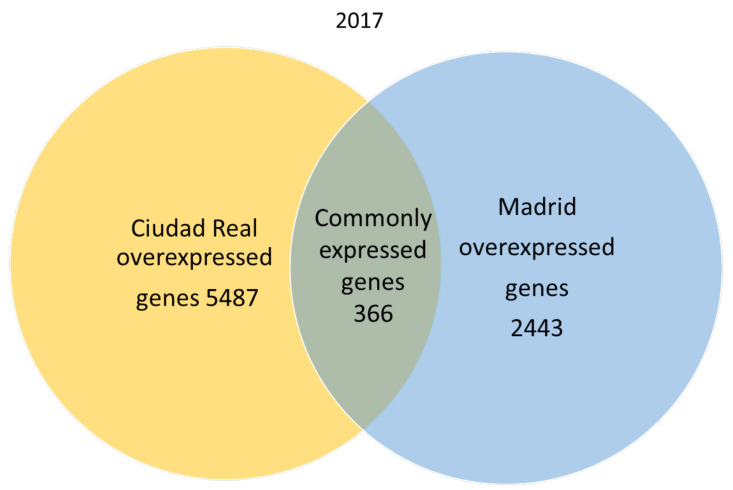
Venn diagram of overexpressed and common genes in the samples of pollen from Madrid and Ciudad Real in 2017 with a *p* adjust value < 0.05.

**Figure 2 plants-09-01507-f002:**
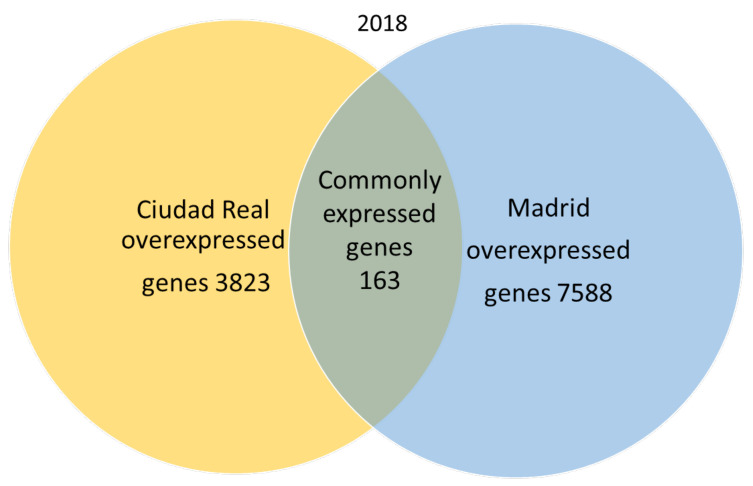
Venn diagram of overexpressed and common genes in the samples of pollen from Madrid and Ciudad Real in 2018 with a *p* adjust value < 0.05.

**Figure 3 plants-09-01507-f003:**
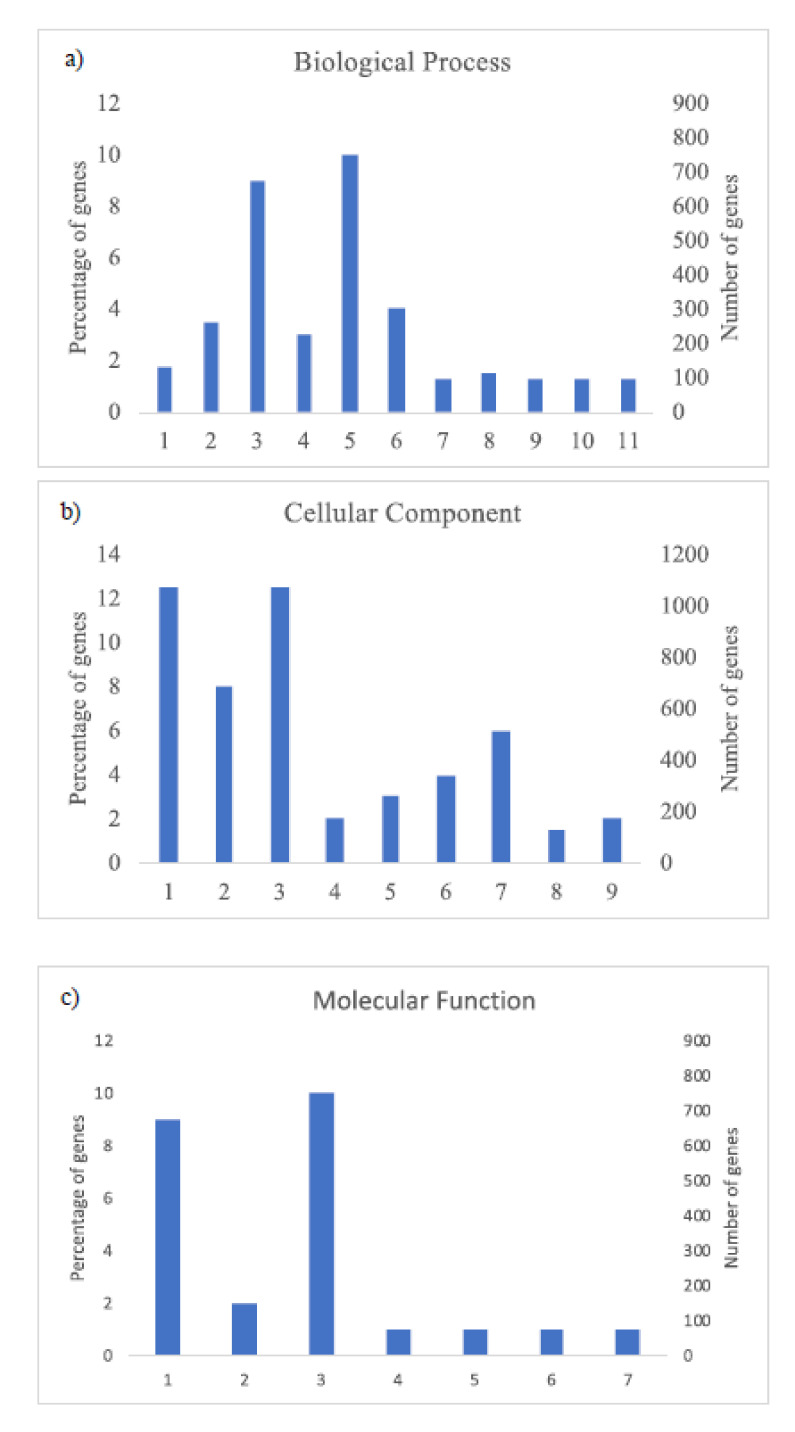
Histogram of GO classifications of *L. perenne* pollen in samples of 2017. Results are summarized for the three main categories: (**a**) Biological process: 1. Localization, 2. Biological regulation, 3. Metabolic process, 4. Regulation of biological process, 5. Cellular process, 6. Response to stimulus, 7. Multi-organism process, 8. Cellular component organization or biogenesis, 9. Developmental process, 10. Multicellular organismal process, 11. Signaling; (**b**) cellular component: 1. Cell part, 2. Organelle, 3. Cell, 4. Protein-containing complex, 5. Organelle part, 6. Membrane part, 7. Membrane, 8. Membrane-enclosed lumen, 9. Extracellular region; and (**c**) molecular function: 1. Catalytic activity, 2. Transporter activity, 3. Binding, 4. Signal transducer activity, 5. Structural molecule activity, 6. Molecular function regulator, 7. Transcription regulator activity.

**Figure 4 plants-09-01507-f004:**
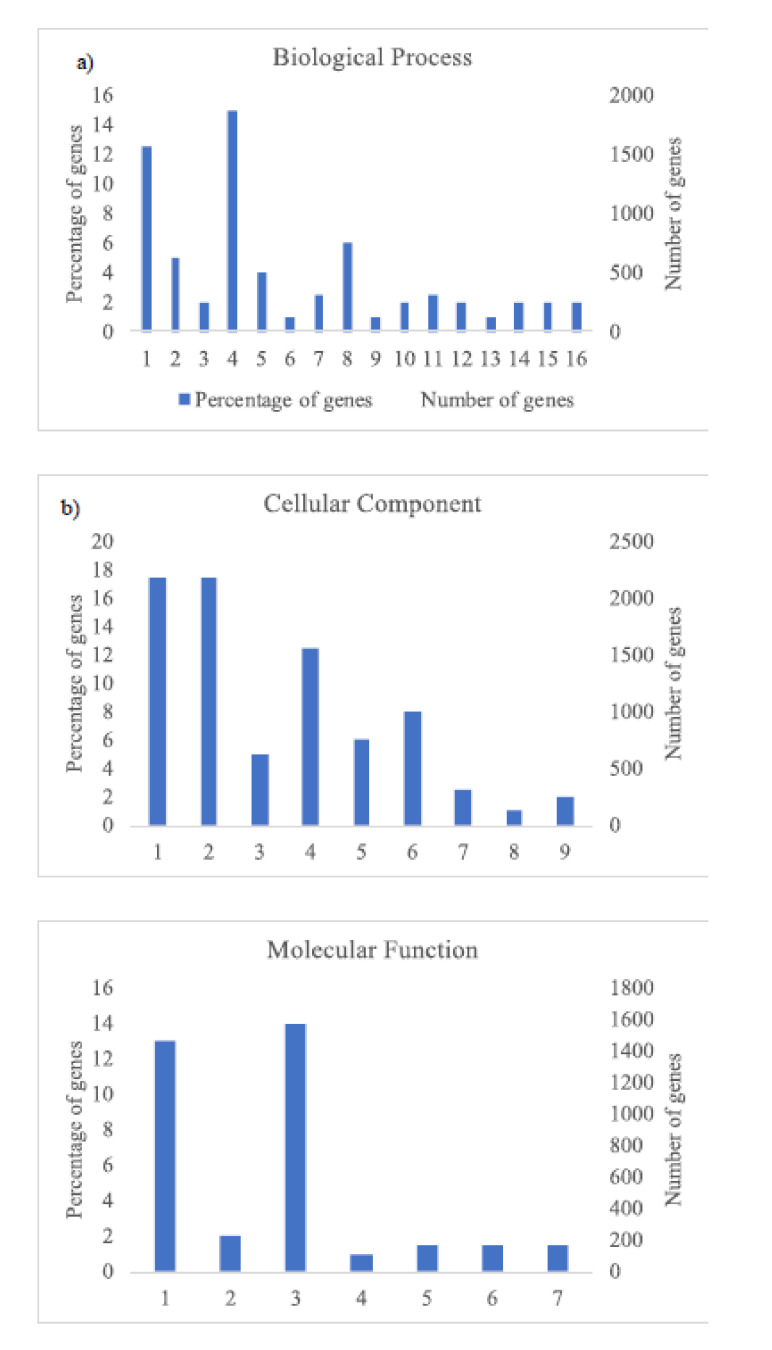
Histogram of GO classifications of *L. perenne* pollen in samples of 2018. Results are summarized for the three main categories: (**a**) Biological process: 1. Metabolic process, 2. Biological regulation, 3. Multi-organism process, 4. Cellular process, 5. Regulation of biological process, 6. Negative regulation of biological process, 7. Localization, 8. Response to stimulus, 9. Immune system process, 10. Signaling, 11. Cellular component organization or biogenesis, 12. Developmental process, 13. Positive regulation of biological process, 14. Multicellular organismal process, 15. Reproductive process, 16. Reproduction; (**b**) cellular component: 1. Cell, 2. Cell part, 3. Organelle part, 4. organelle, 5. Membrane part, 6. Membrane, 7. Protein-containing complex, 8. Membrane-enclosed lumen, 9. Extracellular region; and (**c**) molecular function: 1. Catalytic activity, 2. Transporter activity, 3. Binding, 4. Molecular transducer activity, 5. Molecular function regulator, 6. Structural molecule activity, 7. Transcription regulator activity.

**Table 1 plants-09-01507-t001:** Total, mapped, and HQ (High Quality) reads of the three different replicates from pollen from Madrid and Ciudad Real in the two sampling moments, 2017 and 2018.

2017	Total Reads	Mapped Reads	%Mapped Reads	HQ Reads	%HQ Reads
Pollen Madrid 1	56.829.540	29.508.133	51.92	17.679.876	31.11
Pollen Madrid 2	51.099.090	27.351.198	53.53	15.463.238	30.26
Pollen Madrid 3	54.889.770	29.095.973	53.01	17.484.232	31.85
Pollen Ciudad Real 1	51.263.160	27.121.640	52.91	17.667.246	34.46
Pollen Ciudad Real 2	54.833.392	30.344.055	55.34	19.671.670	35.88
Pollen Ciudad Real 3	44.148.344	23.470.567	53.16	15.174.414	34.37
**2018**	**Total reads**	**Mapped reads**	**%Mapped reads**	**HQ reads**	**%HQ reads**
Pollen Madrid 1	69.632.502	36.876.163	52.96	24.110.014	34.62
Pollen Madrid 2	57.806.190	31.515.221	54.52	21.451.848	37.11
Pollen Madrid 3	70.511.258	38.187.455	54.16	25.894.258	36.72
Pollen Ciudad Real 1	57.131.912	28.713.056	50.26	18.545.118	32.46
Pollen Ciudad Real 2	51.317.428	26.177.323	51.01	16.593.250	32.33
Pollen Ciudad Real 3	51.544.796	26.470.467	51.35	16.975.778	32.93

**Table 2 plants-09-01507-t002:** Among the 50 genes with the highest differential expression (based in the fold change value), those that have some description in the databases used are indicated in this table: GO (Gene Ontology), KO (KEGG Orthology), Pfam, and COG (Clusters of Orthologous Groups). (a) Genes overexpressed in Madrid in 2017; (b) genes overexpressed in Ciudad Real in 2017; (c) genes overexpressed in Madrid in 2018; (d) genes overexpressed in Ciudad Real in 2018. The rest of the genes can be found in the Appendix A.

(**a**)					
**ID.**	**Fold Change**	**GO_Description**	**KO_Definition**	**Pfam_Description**	**COG_Description**
MEHO01012663.1	56.60	F:shikimate O-hydroxycinnamoyltransferase activity		Transferase	
MEHO01021292.1	32.46	F:metal ion binding|F:nucleic acid binding|F:RNA-DNA hybrid ribonuclease activity		zf-RVT	
MEHO01007492.1	24.03	C:cytoplasm|F:DNA binding|F:exonuclease activity|F:metal ion binding|F:RNA binding|F:RNA-DNA hybrid ribonuclease activity|P:DNA replication, removal of RNA primer	ribonuclease HI [EC:3.1.26.4]		Ribonuclease HI
MEHO01503913.1	18.82	C:apoplast|F:guiding stereospecific synthesis activity|P:phenylpropanoid biosynthetic process		Dirigent	
MEHO01031871.1	16.10	F:zinc ion binding		DYW_deaminase|PPR|PPR_1|PPR_2	
MEHO01022721.1	14.80			Exo_endo_phos|RVT_1	
MEHO01032766.1	14.38	C:endoplasmic reticulum|P:response to heat	HSP20 family protein	HSP20	Molecular chaperone IbpA, HSP20 family
MEHO01070426.1	14.35	F:transferase activity, transferring hexosyl groups|P:metabolic process		UDPGT	UDP:flavonoid glycosyltransferase YjiC, YdhE family
(**b**)					
**ID**	**Fold Change**	**GO_Description**	**KO_Definition**	**Pfam_Description**	**COG_Description**
MEHO01015366.1	−115.42	C:integral component of membrane|F:zinc ion binding|P:defense response		Gly-zipper_YMGG|zf-RING_2	
MEHO01137086.1	−104.50	C:nucleus|P:regulation of transcription, DNA-templated|P:transcription, DNA-templated		CCT_2|tify	
(**c**)					
**ID**	**Fold Change**	**GO_Description**	**KO_Definition**	**Pfam_Description**	**COG_Description**
MEHO01214398.1	2789.50	F:lipid binding|P:lipid transport|P:response to ethylene|P:response to hydrogen peroxide|P:response to salicylic acid|P:response to wounding		Tryp_alpha_amyl	
MEHO01067431.1	1478.00	C:cytoplasm|P:protein homooligomerization		HSP20	
MEHO01094758.1	1419.95	C:cytoplasm|P:protein homooligomerization		HSP20	
MEHO01017478.1	1297	C:chloroplast		HSP20	
MEHO01041587.1	1074.98	F:lipid binding|P:lipid transport		Tryp_alpha_amyl	
MEHO01207432.1	1071.85	C:nucleus|P:response to heat	HSP20 family protein	HSP20	Molecular chaperone IbpA, HSP20 family
MEHO01397565.1	950.93	C:cytoplasm|P:protein homooligomerization		HSP20	
MEHO01126086.1	944.40	C:cytoplasm|P:protein homooligomerization		HSP20	
MEHO01035973.1	880.14	C:chloroplast|P:defense response|P:response to biotic stimulus		Bet_v_1	
MEHO01067929.1	862.03	C:cytoplasm		HSP20	
MEHO01121044.1	786.38	C:cytoplasm|P:response to arsenic-containing substance|P:response to cadmium ion|P:response to copper ion|P:response to ethanol|P:response to heat|P:response to hydrogen peroxide	HSP20 family protein	HSP20	Molecular chaperone IbpA, HSP20 family
MEHO01028187.1	766.82			Lir1	
MEHO01022615.1	707.14			DUF4283|zf-CCHC_4	
MEHO01019001.1	689.60	C:cytoplasm		HSP20	
MEHO01005459.1	666.64	C:cytoplasm|P:defense response|P:response to biotic stimulus		Bet_v_1	
MEHO01009825.1	664.78	C:endoplasmic reticulum membrane|F:FK506 binding|F:peptidyl-prolyl cis-trans isomerase activity|P:chaperone-mediated protein folding	FK506-binding protein 1 [EC:5.2.1.8]	FKBP_C	FKBP-type peptidyl-prolyl cis-trans isomerase
MEHO01532070.1	613.47	C:cytoplasm		HSP20	
MEHO01126952.1	547.02	C:nucleus|P:response to heat	HSP20 family protein	HSP20	Molecular chaperone IbpA, HSP20 family
MEHO01340596.1	545.97	C:nucleus|P:response to heat	HSP20 family protein	HSP20	Molecular chaperone IbpA, HSP20 family
MEHO01043840.1	462.16	F:metal ion binding|F:nucleic acid binding|F:RNA-DNA hybrid ribonuclease activity		zf-RVT	
MEHO01120068.1	44.88	C:apoplast|C:cell wall|F:sucrose alpha-glucosidase activity|P:carbohydrate metabolic process	beta-fructofuranosidase [EC:3.2.1.26]	Glyco_hydro_32C|Glyco_hydro_32N	Sucrose-6-phosphate hydrolase SacC, GH32 family
MEHO01032766.1	439.34	C:endoplasmic reticulum|P:response to heat	HSP20 family protein	HSP20	Molecular chaperone IbpA, HSP20 family
MEHO01089025.1	427.45	F:lipid binding|P:lipid transport		Tryp_alpha_amyl	
MEHO01005917.1	348.32			F-box|FBD|LRR_2	
MEHO01208798.1	346.28	C:mitochondrion	HSP20 family protein	HSP20	
(**d**)					
**ID**	**Fold Change**	**GO_Description**	**KO_Definition**	**Pfam_Description**	**COG_Description**
MEHO01067659.1	−425.58	C:integral component of membrane|F:transporter activity|P:nitrate assimilation		PTR2	Dipeptide/tripeptide permease
MEHO01000853.1	−320.23	F:metal ion binding|F:nucleic acid binding|F:RNA-DNA hybrid ribonuclease activity		zf-RVT	
MEHO01015099.1	−320.18			Exo_endo_phos|RVT_1	
MEHO01121030.1	−241.91	C:extracellular region|C:membrane|F:acid phosphatase activity|F:metal ion binding		Metallophos	3’,5’-cyclic AMP phosphodiesterase CpdA
MEHO01357234.1	−233.31	F:aspartic-type endopeptidase activity|F:endonuclease activity|F:nucleic acid binding|F:RNA-directed DNA polymerase activity|F:zinc ion binding|P:DNA integration		gag_pre-integrs|rve|RVT_2|zf-CCHC	
MEHO01008671.1	−201.64	C:integral component of membrane|C:plasma membrane|F:dipeptide transporter activity|P:dipeptide transport|P:pollen tube growth|P:protein transport		PTR2	Dipeptide/tripeptide permease
MEHO01111302.1	−172.68	C:mitochondrion		RVT_2	Transposase InsO and inactivated derivatives
MEHO01012083.1	−166.71	C:integral component of membrane|C:plasma membrane|F:transmembrane transporter activity|P:transport		EamA	

**Table 3 plants-09-01507-t003:** Protein encoded by the overexpressed genes in Madrid pollen samples.

Protein Encoded by the Overexpressed Genes	Number of Isoforms in 2017	Number of Isoforms in 2018
HSP (20, 70, and 90)	47	60
Glycoside hydrolase	25	42
Leucine-rich repeat	12	38
EF-Hand family	9	13
Pollen allergen 1	5	4
Cofilin	5	2
Pectinesterase	7	
Serpin	5	
Lipid transfer proteins		7
Thaumatin		6

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
