# Peer review of "Search for New Allergens in Lolium perenne Pollen Growing under Different Air Pollution Conditions by Comparative Transcriptome Study"

_plants, 2020, doi:10.3390/plants9111507_

Round 1

Reviewer 1 Report

This manuscript attempts to describes efforts to identify pollution induced pollen allergens using a transcriptomic approach.

Unfortunately, the manuscript has not been prepared well enough for me to make a good assessment of the quality of the science presented, and it needs extensive language editing before such a judgement can be made.

I do sympathize with researchers who have to prepare manuscripts in a second language – but the problems here go beyond just language. Despite having downloaded the template document that Plants kindly and unusually makes available, this manuscript ignores some of the major instructions – for example, that the references should be numbered and that methods should be at the end in section 4 (not 2). Beyond that there are many typographical (not just grammatical) errors throughout the text.

In addition. no effort has been made to check that the figures are legible or even complete – figures 1 and 2 have a lot of text labels that are unreadable, figures 3 and 4 claim to be Venn diagrams, but are text only.

I was also surprised to see the statement “All authors contributed equally to this work”. This is the first time I have ever seen this. Having read the Author Contribution section, it does not seem valid, “All authors are integrated into the research project PI15/00. All of them have contributed equally to the design of the experiment. JAL and EGM have been in charge of interpreting the results obtained from the RNAseq. J.A.L. and EGM redacted the manuscript.”

Based on this statement it seems that it would be more appropriate to have just J.A.L. and EGM as joint first authors………

I appreciate that authorship might be different in medical and/or pharmacy fields, but the people who actually do the work get the most credit in plant science!

If this work was language edited and presented clearly with legible figures, it might be acceptable for Plants, but in its current state I cannot say this with certainty, attention to detail is critical in transcriptome analyses, and given the state of the manuscript submitted I have major concerns!

Author Response

First of all, we would like to thank you for your constructive criticisms, which have substantially improved the quality of the manuscript. All the comments and suggestions raised by reviewers about the manuscript have been addressed in the same order in italics:

Reviewer 1

This manuscript attempts to describes efforts to identify pollution induced pollen allergens using a transcriptomic approach.

Unfortunately, the manuscript has not been prepared well enough for me to make a good assessment of the quality of the science presented, and it needs extensive language editing before such a judgement can be made.

The manuscript has been revised by a native English-speaking colleague. All modifications have been indicated in the track changes of Microsoft Word.

I do sympathize with researchers who have to prepare manuscripts in a second language – but the problems here go beyond just language. Despite having downloaded the template document that Plants kindly and unusually makes available, this manuscript ignores some of the major instructions – for example, that the references should be numbered and that methods should be at the end in section 4 (not 2). Beyond that there are many typographical (not just grammatical) errors throughout the text.

Regarding the comment about ignoring some of the author's instructions, I can only apologize. The reviewer is right and I cannot make excuses. I can only thank you that despite this you have positively assessed the scientific contribution of the manuscript. The manuscript has been reformatting following the author´s information.

In addition. no effort has been made to check that the figures are legible or even complete – figures 1 and 2 have a lot of text labels that are unreadable, figures 3 and 4 claim to be Venn diagrams, but are text only.

The reviewer is right, figures 1 and 2 do not look good despite efforts to improve their quality. On the other hand, these figures provided little information without a study of the genes that were reflected in them. We have changed these figures for a table where the 50 genes included in the figures are reflected. In this table, only those genes that have some description about their function in the consulted databases are shown: Gene Ontology (GO), KEGG Orthology (KO), Pfam and Clusters (COG). Complete information on all genes, common and differentially expressed in each population, can be consulted in the supplementary material provided.

Regarding Venn diagrams, it is very strange because they look good on my computer (I work with Mac) but when the document is opened in windows, only the letters in the figure are visible. We have modified some things in the diagrams and we think that now they will look good.

I was also surprised to see the statement “All authors contributed equally to this work”. This is the first time I have ever seen this. Having read the Author Contribution section, it does not seem valid, “All authors are integrated into the research project PI15/00. All of them have contributed equally to the design of the experiment. JAL and EGM have been in charge of interpreting the results obtained from the RNAseq. J.A.L. and EGM redacted the manuscript.”

Based on this statement it seems that it would be more appropriate to have just J.A.L. and EGM as joint first authors………

It is certainly an unfortunate phrase; it does not absolutely reflect the contribution of each author. As the reviewer knows well, a manuscript is the last step in a long way in an investigation. The manuscript is the result of much previous work. What we wanted to indicate is that throughout all this work all the authors have had a similar contribution. We have modified that paragraph to clarify it and make the contribution of each author more explicit.

I appreciate that authorship might be different in medical and/or pharmacy fields, but the people who actually do the work get the most credit in plant science!

  The contribution of the authors has been rewritten to better reflect what each of them contributed. Regarding the claim that all authors have contributed equally to the manuscript, the reviewer is right that it is not justified based on what is written in the paragraph. As the reviewer knows, publishing a manuscript takes a lot of work, design work meetings, etc. We tried to reflect all that work and effort in that phrase. But I agree that as that paragraph was written, this explanation could not be understood. But the truth is that if we value all that effort, we have all contributed equally.

If this work was language edited and presented clearly with legible figures, it might be acceptable for Plants, but in its current state I cannot say this with certainty, attention to detail is critical in transcriptome analyses, and given the state of the manuscript submitted I have major concerns!

We have made a thorough revision of the writing of the manuscript, in addition we have suppressed Figures 1 and 2 to replace them with a table with more relevant information, and more details about RNAseq data has been included in results section. On the other hand, we have introduced several paragraphs in the introduction to further clarify the objective of the work. I hope that with all these changes the revierwer feels satisfied and considers it appropriate that the manuscript can be published in Plants.

Reviewer 2 Report

Dear Authors,

My review is attached as a pdf file.

Reviewer

Author Response

First of all, we would like to thank you for your constructive criticisms, which have substantially improved the quality of the manuscript. All the comments and suggestions raised by reviewers about the manuscript have been addressed in the same order in italics:

Reviewer 2

Abstract: “The population of Madrid with higher air pollution conditions.” Please clarify this

sentence.

We have clarify this sentence introducing a new phrase: “The populations of L. perenne in Madrid develop under conditions of greater air pollution, supporting higher levels of NO2 and SO2, than the populations of Ciudad Real.

line 59: correctly: “higher atmospheric pollution”, instead of “more atmospheric pollution”;

DONE

line 61: correctly: “a greater degree”, instead of “an greater degree”;

DONE

line 61: correctly: “All these were reflected”, instead of “All this was reflected”;

DONE

Figure 3 and Figure 4 seem for me unfinished / incomplete. Please check them!

Figure 5 and Figure 6: these figures are found on two and three pages, respectively. This is not a fortunate solution. Instead, I suggest to find another way compressing the four sub-figures onto one page. A possibility: the name of the columns below the sub-figures would get numbers, and the text of these numbers would be structured in the text of Figure 5 and Figure 6, respectively.

Regarding Venn diagrams (Figs 3 and 4; 1 and 2 in revised manuscript), it is very strange because they look good on my computer (I work with Mac) but when the document is opened in windows, only the letters in the figure are visible. We have modified some things in the diagrams and we think that now they will look good.

Figures 5 and 6 (3 and 4 in revised manuscript) have been modified following the suggestions of the reviewer.

ï‚· lines 309-310: What do you think, what might be the reason that some proteins are overexpressed in one year but not in the following year? Internal or external reasons? This point might be a subject of another research.

The metabolism of plants is very complex. In fact, it is divided into primary and secondary. The secondary, also called adaptive, is used by plants to relate to their environment and adapt to its changes. Unlike primary metabolism, secondary metabolism is an inducible metabolism, it is activated under different stresses, in a very specific way. This means that any environmental variation (biotic or abiotic) can be perceived by the plant and can trigger different routes of said adaptive metabolism, generating molecules of different chemical nature. To the question asked by the reviewer, I believe that external reasons have a greater influence on the fact that some proteins are overexpressed one year and not another. In both years of sampling, the values ​​of NO2, SO2 and other air pollutants did not have significant variations, but there were variations in other values ​​such as precipitation and temperature. In the month of May 2017 in Madrid the precipitation was 27.1 mm and there was an average temperature of 20.1ºC, while in 2018 there was rainfall of 78.4 mm and an average temperature of 17.1ºC. Those differences are sure to have an important influence. For this reason, in the work we have focused on proteins overexpressed for two years and with a high number of isoforms, although as can be seen in the complementary material provided, there are some proteins that are overexpressed for only one year or with few isoforms, which could be very interesting to study. As the reviewer says, the work continues. We have asked for more funding to continue deepening even more since it is a very worrying public health problem and research is the way to improve.

Reviewer 3 Report

The manuscript about searching for new allergens in L. perenne needs a major revision. First of all, the RNASeq study does not have any control. Both Madrid and Ciudad Real populations are your experimental samples, so to compare both samples, you should be having a control and then compare control vs experimental sample and make comparisons between the two experimental samples LIMMA is a powerful tool that allows you compare direct effects and interaction effects as well.

For claiming your differentially expressed genes to be allergenic, some of the protein domain prediction studies are necessary.

No comparison is done with already known allergen genes in different grasses.

No information given about biological replicates being assessed in the methods.

If you are using same population of L.perenne, there is no point of why you get so less mapping percentages. Have you checked what the source of contamination in your samples is. No trimming performed on raw reads, what quality reads were?

Raw data has not been submitted to SRA.

How many allergen genes are there in L. perenne pollen already known – not specified anywhere.

Figure 1, 2 – Very old fashioned heatmap presentation, many R libraries available to make heatmaps more presentable.

Line18: increasing this....avoid using pronouns and be more specific.

Line 20: How many pollen genes were considered for the study and how many are known as allergenic?

Line 22: Only genes are differentially expressed not protein. This needs updation in the entire manuscript.

Line 153: 666,181 genes – are these genes expressed in pollen?

Table 2: Name of the genes mentioned in the table are not given.

Author Response

First of all, we would like to thank you for your constructive criticisms, which have substantially improved the quality of the manuscript. All the comments and suggestions raised by reviewers about the manuscript have been addressed in the same order in italics:

Reviewer 3

The manuscript about searching for new allergens in L. perenne needs a major revision. First of all, the RNASeq study does not have any control. Both Madrid and Ciudad Real populations are your experimental samples, so to compare both samples, you should be having a control and then compare control vs experimental sample and make comparisons between the two experimental samples LIMMA is a powerful tool that allows you compare direct effects and interaction effects as well.

I do not agree with your assessment. This work derives from a research project and another published paper in which we demonstrate a greater allergenic capacity in pollen produced by L. perenne plants in Madrid than in pollen from plants growing in Ciudad real (Lucas et al., 2019; Plant Physiology and Biochemistry, 135: 331-340). Furthermore, the two allergists who sign on the manuscript have verified this difference in their consultations. There are many publications that relate air pollution with a greater allergenic capacity of pollen. Our control is the pollen of Ciudad Real, a city with little air pollution.

For claiming your differentially expressed genes to be allergenic, some of the protein domain prediction studies are necessary.

I agree, but I think that nowhere have I stated what the reviewer says. The work tries to identify possible new allergens. We have asked for more funding to continue studying, and to be able to affirm that some of the encoded proteins for the genes overexpressed  found in this work are new allergens. The work is very novel and original, and for the first time in the literature there is talk of possible new allergens in L. perenne, one of the most ubiquitous allergenic plants.

No comparison is done with already known allergen genes in different grasses.

In the discussion we have tried to find allergens in other plants that coincide with those that have been overexpressed in our work on L. perenne. In most cases we have found allergens already described in other plants that coincide with the possible new allergens of L. perenne. This is important, because it reinforces our hypothesis that it is very likely that in a short time we will be able to affirm this fact without a doubt.

No information given about biological replicates being assessed in the methods.

If you are using same population of L.perenne, there is no point of why you get so less mapping percentages. Have you checked what the source of contamination in your samples is. No trimming performed on raw reads, what quality reads were?

In the table where the quality data of the RNAseq results are indicated, I do not see that there are great differences in the percentage of mapping. In the report that the company that made the RNAseq gave us and in conversations with them, they confirmed that the results are of very good quality.

Raw data has not been submitted to SRA.

In complementary material we have uploaded the sheets excell where are all the data provided by the company that did the RNAseq analysis. In these excel leaves are all the genes analyzed without removing absolutely nothing.

How many allergen genes are there in L. perenne pollen already known – not specified anywhere.

The allergens described in L. perenne today are indicated in the discussion.

Figure 1, 2 – Very old fashioned heatmap presentation, many R libraries available to make heatmaps more presentable.

Figures 1 and 2 have been changed by a table, this same assessment has been made by another of the referees. We agree that they were not of good quality and also provided little information. We believe that the table and its explanation provide more relevant information.

Line18: increasing this....avoid using pronouns and be more specific.

The work has been deeply grammatically corrected.

Line 20: How many pollen genes were considered for the study and how many are known as allergenic?

All this information has been indicated in the manuscript. We believed that it was not necessary in the abstract, but we can incorporate it if the reviewer considers it essential.

Line 22: Only genes are differentially expressed not protein. This needs updation in the entire manuscript.

Indeed, this has been corrected in the review process.

Line 153: 666,181 genes – are these genes expressed in pollen?

The reviewer is right that this information on the number of genes was not well explained in the manuscript, despite all the information being in the supplementary material. We have added more information about it in the results section.

Table 2: Name of the genes mentioned in the table are not given.

It is true, in the table are the proteins that, according to the Pfam databases, are encoded by the overexpressed genes. We have modified the explanation of the table to correct it.

Round 2

Reviewer 1 Report

The authors have made good effort to address my previous comments, as well as the other reviewers and the manuscript is considerably improved.

It is clear to me that there us valuable information in this manuscript (which is not always the case with RNAseq studies!). In an ideal world it would be helpful to have empirical data on whether the actual pollen sampled actually differs in allergenic capacity, which might in particular help in assessing the relevance of differences between years in the genes identified. That being said I do judge the RNAseq data to be controlled and possess the 3 biological replicate minimum that is generally required these days - albeit in a slightly unusual way in that replicate samples were taken from populations growing close together.

I appreciate that this is a work in progress and that the intention is to identify genes on which to focus future studies to pin down the exact genes causing allergies.  

I thank the authors for clarifying the authorship statement and roles each author played - this is much clearer now. I also appreciate the substantial improvements in formatting, figures and grammar.

The manuscript has been much more carefully prepared and is much clearer now, that being said, there are still a substantial number of grammatical issues and the manuscript would benefit substantially from being professionally language edited.

Author Response

First of all, I want to thank the reviewer for his kind words about the work, and for all the improvements he has made. Undoubtedly, the work has improved a lot after your contributions.

The clinical groups that are working with the same pollen of the work, are doing tests in vitro and in vivo to verify the differences of allergenicity between them. They have not finished yet, but they have already told me that the first results indicate a greater allergenicity in pollen from Madrid than from Ciudad Real.

It is true that the populations chosen for the sampling are not very far from each other, but we tried to ensure that they were separated from each other by some physical separation (a street, a sidewalk, etc.), so it cannot be strictly stated that they develop at the same place.

We have done our best effort to improve english style and grammar. The manuscript has been revised by a scientific colleague who collaborates with the department of modern languages ​​of the University helping the researchers in the writing of the articles. We hope that after this revision, the grammatical and linguistic problems that the manuscript had have been solved. However, we would be glad to incorporate more specific suggestions if necessary.